# Quinoa (Chenopodium Quinoa Willd.) as Functional Ingredient for the Formulation of Gluten-Free Shortbreads

**DOI:** 10.3390/foods13030377

**Published:** 2024-01-24

**Authors:** Elisabetta Bravi, Valeria Sileoni, Ombretta Marconi

**Affiliations:** 1Italian Brewing Research Centre, University of Perugia, Via San Costanzo ncn, 06126 Perugia, Italy; elisabetta.bravi@unipg.it; 2Department of Economic and Legal Sciences, Universitas Mercatorum, Piazza Mattei 10, 00186 Rome, Italy; valeria.sileoni@unimercatorum.it; 3Department of Agricultural, Food and Environmental Sciences, University of Perugia, Borgo XX Giugno 74, 06121 Perugia, Italy

**Keywords:** gluten-free, shortbreads, quinoa flour, rice flour, amino acids, antioxidant activity, fiber

## Abstract

The incidence of celiac disease and gluten intolerance has been significantly rising globally. Gluten-free product consumption registered a sudden rise also among tolerant people, due to psychosocial factors. Biscuits are popular, low-cost bakery foods, consumed by nearly everyone worldwide. The removal of gluten from the baked product causes some undesirable traits and different textures and tastes. The main goal consists in creating a food product with the same taste and texture as a product with gluten. Moreover, gluten-free bakery products are usually low-grade sources of protein and poor in dietary fiber. Quinoa is a source of total dietary fiber and valuable protein. In this study, quinoa flour was used as the main constituent in the formulation of gluten-free shortbreads to improve their nutritional properties. Six different recipes with different percentages of quinoa flour have been realized. The formulations were compared with each other and with a wheat flour control shortbread, using textural analysis. The experimental biscuits with textural features more similar to control shortbread were subjected to a triangle-discriminating and preference test and those selected by panelists was characterized from a chemical-physical and sensorial point of view. The experimental shortbreads constituted a good compromise to exploit the good nutritional composition of quinoa while maintaining an acceptable sensory profile.

## 1. Introduction

Quinoa (*Chenopodium quinoa* Willd.) is an ancient pseudocereal of the Amaranthaceae family, mainly grown in the Andean region but actually cultivated on all continents. Quinoa plants can resist freezing temperatures as well as drought [1], so they can adapt to different agro-environmental conditions and can be cultivated almost everywhere. Quinoa is an annual herbaceous plant differently colored (yellow, purple, red, or green) depending on the variety, characterized by the presence of protein with high biological value, and rich in both essential and non-essential amino acids and dietary fiber. Quinoa is considered a valuable source of vegetal protein, with a protein content comparable to milk and higher than the protein levels of other cereals such as wheat, rice, and maize [2].

Moreover, high levels of essential fatty acids, vitamins, and minerals are present in quinoa, and it is a gluten-free (GF) raw material suitable for celiac consumers [2]. The interest in quinoa has been increasing in the past few years, and this pseudocereal has been called “golden grain” due to its high nutritional value [3]. The FAO recognizes quinoa for its nutritional properties and its genetic diversity, as well as the cultural and socio-economic benefits it has on the local environment [4]. Furthermore, NASA selected quinoa as an ideal diet for its astronauts on board space missions [5]. Additionally, the cultivation and consumption of quinoa have steadily increased worldwide thanks not only to its dietary features but also to its capacity to tolerate hostile conditions and its ability to adapt to very different environmental situations [3]. 

Since 2015, ISTAT has included GF biscuits and pasta in the basket, recommending the regular consumption of these products even among tolerant consumers, and, gradually, the interest in GF products, with proper characteristics similar to conventional products, is growing [3]. Globally, in the Western area, the incidence of celiac disease and gluten intolerance has been increasing since the second half of the 1990s. In other regions, such as Africa, Asia, and South America, more research is necessary to understand the real incidence of this kind of disease [6]. Thereafter, GF products registered a sudden rise in the global market and this increase was observed even among tolerant people. The reasons for this phenomenon appear to be complex because of the different factors involved, such as consumers’ choices, product features, and psychosocial aspects. Moreover, the general belief that GF bakery products are healthier than traditional ones, because they are known to have a low glycemic index and low calorie content, and help to control weight and fight obesity, has driven a further rise in demand [6,7]. 

In this global context, the need for new valuable GF products has increased, and nowadays, the main goal consists in creating a food product with the same texture and taste as a product with gluten, so that the consumer cannot tell the difference [7]. The removal of gluten from the baked product causes some undesirable traits in the final product such as grittiness, low volume, and a reduced shelf life. Nevertheless, by using the right grain and granulation, it is possible to solve the grittiness issue and provide an appealing visual element to the product. Nevertheless, the primary issue is texture; it is very challenging to create the right one without gluten [7]. 

Biscuits are one of the most widespread low-cost bakery foods, attractive to consumers of all ages (including the elderly). These products are ready-to-eat and easily available in a wide variety of shapes, sizes, tastes, and packs, with a long shelf life due to the low moisture content (<5%). Moreover, they are an important source of energy [8,9]. Several efforts have been made to modify biscuits’ composition to increase their nutritive value. Usually, a conventional strategy to reach this goal is increasing the content of dietary fiber and protein in the final product, to attend to the consumers’ growing demand for functional food [10]. Furthermore, GF bakery products usually have a low content of dietary fiber, because of the use of pure refined flour or starch as primary raw materials in their formulation, and low-grade sources of protein. This gives rise to the need to improve not only the quality but also the nutritional profile of GF products. For this reason, the food industry has an interest in pushing research to explore new raw materials and technologies for producing GF bakery products. Rice flour (RF) is the most appropriate ingredient for GF bakery formulations because of its features, such as high digestibility, white color, hypoallergenic properties, and neutral taste [10].

The aim of this study, structured on the basic idea that the use of quinoa flour (QF) could not only satisfy the needs of gluten deprivation but also the nutritional ones, concerned the formulation of a GF shortbread that could satisfy the expectations of the consumer, whether celiac or not, and at the same time offer a nutritionally interesting composition. On the other hand, RF, naturally GF, was hypothesized to be useful for improving the taste by attenuating the typical aroma of quinoa. In this research, a shortbread with QF as the main constituent was formulated. RF was added to give a more neutral flavor, keeping the product naturally GF. Chemical-physical analyses (moisture, ash, aw, pH, carbohydrates, simple sugars, fats, fatty acids, proteins, amino acids, fiber, polyphenols, and antioxidant assays) were carried out on raw materials, wheat flour (WF), QF, and RF. After that, six different recipes with different percentages of QF (from 10 to 100%) and RF were realized. The formulations were compared with each other and with a conventional shortbread (control shortbread, CS) made up of 100% WF, using textural analysis. The recipes technologically more similar to CS were selected and tested with a difference/preference analysis. From the obtained results, the shortbread with 60% quinoa was selected as being the more similar to CS, and the one most appreciated in the sensory analysis. In the next step, the quinoa shortbread preferred by the panelists and the CS were characterized from a chemical-physical and sensorial point of view. From a nutritional point of view, the experimental shortbread, was rich in essential amino acids, with a good composition of unsaturated fatty acids, and a good content of fiber and antioxidants. Moreover, the sensory analysis was overall positively assessed by the panelists.

## 2. Materials and Methods

### 2.1. Materials

#### Shortbreads Ingredients

WF (“00”, with 11% of moisture, 9% of protein, 0.57% of ash, and a strength (W) of 146) was acquired from Molini Popolari Riuniti di Ellera-Umbertide S.a.c.r.l., (Perugia, Italy). White biological and stone-ground QF (variety Qhaslala, non-EU origin) with reduced saponin content and white biological RF (Italian origin) were purchased from Molino Agostini S.r.l (Perugia, Italy). The other shortbread raw materials were sucrose (21.2%, Eridania Italia S.p.A, Bologna, Italy), butter (15.2%, Grifolatte, Perugia, Italy), water (4.5%), class A eggs (9.2%, Ovito, Perugia, Italy), salt (0.2%, Italkali, Palermo, Italy), sodium bicarbonate (0.2%, E 500; Solvay, Milano, Italy), and ammonium bicarbonate (0.2%, E 503; Bertolini, Brescia, Italy).

### 2.2. Methods

#### 2.2.1. Experimental Design

The entire experimental design is summarized in the scheme in Figure 1. The WF, QF, and RF were analyzed for the following parameters: water activity (aw), moisture, ash, total nitrogen and proteins, fat, carbohydrates, total dietary fiber (TDF), free amino acids (AAs), fatty acids (FAs), polyphenol content, and antioxidant capacity. Then, seven different shortbreads were prepared: the CS made up of WF and six experimental shortbreads formulated with QF and RF at different percentages of quinoa flour (from 0 to 100%, named Q0S, Q10S, Q20S, Q40S, Q60S, and Q100S). The different formulations of shortbreads were evaluated for their textural properties, and in particular for hardness. In the next step, the recipes technologically most similar to the conventional one were selected and subjected to a triangle-discriminating and preference test. Finally, the experimental shortbread that the panelists preferred was characterized from a chemical-physical and sensorial point of view and compared with the CS.

For the shortbread evaluation, the following parameters were considered: aw, moisture, ash, proteins, fat, carbohydrates, TDF, phenol content, antioxidant capacity, free AAs, FA profile, spread factor, and weight loss. 

The experimental shortbreads were evaluated using sensory analysis, in particular quantitative descriptive analysis, to evaluate the statistically significant difference between the CS and the preferred experimental shortbread.

#### 2.2.2. Analytical Determination

All the analytical determinations were performed in duplicate. 

Water content, ash, pH, and protein contents were determined according to their respective Association of Official Analytical Chemists (AOAC) methods [11]. The nitrogen conversion factors 5.7 and 6.25 were used for the calculation of the protein content of flour and shortbread samples, respectively [12].

An aw meter (Aqualab^®^ series 3, Decagon Devices, Inc., Pullman, Washington, DC, USA), calibrated with lithium chloride solution (aw = 0.250 ± 0.003), was used to determine the aw of the raw materials and shortbread samples. 

A Megazyme assay kit (Megazyme International Ireland, Wicklow, Ireland), was used to determine the TDF content [13].

Carbohydrates were calculated as the percentage difference with moisture, proteins, ashes, fats, and TDF [14].

Free sugars were extracted by using an 80% ethanol solution and then analyzed using a high-performance liquid chromatography—evaporative light scattering detector (HPLC-ELSD) [11]. A Jasco PU-lS80 pump (Jasco Corporation, Tokyo, Japan) performed the gradient through the low-pressure mixing of acetonitrile and water. The flow rate was 1 mL/min at room temperature. The eluent mixture was maintained at 75% (*v*/*v*) acetonitrile in water for 10 min after injection, then programmed to 50% (*v*/*v*) acetonitrile/water for 15 min, and finally maintained at this concentration for 5 min. The separation was performed by using a polymeric amino column Shodex- NH2P-50, 4.5 × 250 mm, (Shodex Inc., Tokyo, Japan). Free sugars were detected using an Evaporative Light Scattering (ELS) Detector C-650 (Buchi Corporation, New Castle, DE, USA) The drift tube temperature was set at 110 °C, and nitrogen flow was set at 2.2 mL/min, gain 10. 

The energy values of shortbreads were calculated using the conversion factors approved by the Food and Agriculture Organization of the United Nations (FAO) (4.0 kcal/g for protein, 9.0 kcal/g for fat, and 4.0 kcal/g for carbohydrates) [15].

The total polyphenol content and antioxidant capacity were determined on flour and shortbread extracts obtained by homogenizing 1 g of sample in 5 mL solution of methanol/water/hydrochloric acid (70:28:2; *v*/*v*/*v*) using a homogenizer (Ultra-Turrax T25, Ika Works Inc., Wilmington, NC, USA) until uniform consistency (2 min). After centrifugation (10 min at 3000 rpm), the supernatants were recovered. The extraction was repeated twice, and the extracts were taken to 10 mL into a volumetric flask. The total polyphenol content was determined by using the Folin–Ciocalteau method [14]. The solution, obtained by adding Folin–Ciocalteu reagent (2 mL) and Na_2_CO_3_ (1.6 mL) to the sample (0.4 mL), was allowed to stay for 2 h at room temperature avoiding the light. The absorbance at 760 nm was finally measured (U-3010/3310, UV/VIS Spectrophotometer, Hitachi, Tokyo) and the total polyphenol content expressed as mg GAE g^−1^ dm (equivalent of gallic acid per g of sample, dry matter) [14]. 

FRAP, ABTS, and DPPH tests were used to measure the antioxidant ability of flour and shortbread samples following the same procedures used by Bravi et al. [14]. Trolox was used as the standard for the calibration curves, and the antioxidant activities measured with the tests were reported as TE g^−1^ dm (Trolox equivalents per g of sample, dry matter). 

Free AA determination was achieved using HPLC-FLD (Fluorimeter Detector). A total of 10 mL of 5% trichloroacetic acid was added to 1 g of flour or ground shortbread and extracted for 30 min under magnetic stirring. The sample, diluted ten times, was filtered with a syringe filter (0.45 μm), derivatized with O-phtaldialdehyde (5 g/L in a ratio of 1 to 2 with the sample), and injected into an HPLC system consisting of a Kinetex EVO C18 column (5 μ, 150 × 4.6 mm, Phenomenex, Torrance, CA, USA). The separation was carried out at 30 °C with a flow rate of 1 mL/min. Mobile phase A was potassium hydrogen phosphate (0.05 M, pH 7.5) and mobile phase B was methanol. The chromatographic separation was achieved in 45 min using the following elution gradient: mobile phase A 81% (0 min), 78% (6 min), 67% (7 min), 56% (30 min), 45% (32 min), 35% (40 min), 81% (42), and 81% (45 min). The detector was an Agilent 1200 fluorescence detector, with excitation/emission wavelengths at 338/420 nm (Agilent Technologies, Santa Clara, CA, USA). The external standard method was used for the calibration, and the calibration plots were constructed for standard compounds with a linearity between 0.1 and 1.7 µg mL^−1^. The AAs detected were aspartic acid, glutamic acid, asparagine, serine, glutamine, histidine, arginine, glycine, threonine, alanine, tyrosine, methionine, tryptophan, valine, phenylalanine, isoleucine, leucine, and lysine. 

The fat extract was used for the determination of FA profiles using gas chromatography. The lipid extracts were trans-esterified through treatment with methanol/KOH solution, and the resulting fatty acid methyl esters were injected into the gas chromatography—flame ionization detector (GC-FID) system (Agilent 6850 GC-FID, Agilent Technologies, Santa Clara, CA, USA, equipped with a DB-23 column (60 m × 0.25 mm × 0.25 μm, Agilent Technologies, Santa Clara, CA, USA). The carrier gas was hydrogen (H_2_) and the flow rate was 1.7 mL/min. The temperatures of the injector and detector were 270 and 280 °C, respectively. The programmed oven temperature was 130 °C for 1 min, raised from 130 to 170 °C at 6.5 °C min^−^^1^, raised from 170 to 215 °C at 2.75 °C min^−^^1^, 215 °C for 12 min, raised from 215 to 230 °C at 40 °C min^−^^1^, and held at 230 °C for 3 min. Peak areas were measured by using an Agilent MSD Chemstation. The FAs were identified by comparing their retention times with those of commercial standards [16].

#### 2.2.3. Laboratory-Scale Shortbread Production

In Table 1, the raw materials of CS and experimental shortbreads with different percentages of QF and RF replacing the WF fraction are shown. To prepare the shortbreads studied in this research, the “doughing up” method was used, as reported by Sileoni et al. [12]. Briefly, a mixture of butter and flour was primarily prepared in a laboratory mixer (KMM770 Chef Major Premier Mixer 1200 W, Kenwood Corporation, Tokyo, Japan) (5 min at 120 rpm and 25 °C). Meanwhile, a second mixture was obtained by mixing the sucrose, the water, the yolk, and the egg white. Then, the two mixtures were combined, and NaCl and the baking powder ((NH_4_)HCO_3_ and NaHCO_3_) were added. The dough was allowed to rest at 4 °C for 1 h, sheeted, and cut into pieces of 40 mm in diameter × 4 mm in thickness. The shortbreads were cooked at 180 °C for 15 min in a professional oven (Combimaster plus 061, Rational AG, Landsberg Am Lech, Germany). The shortbreads were then cooled and packaged in polypropylene bags. 

#### 2.2.4. Shortbread Texture

The hardness was determined on all formulations of shortbread; all samples were analyzed ten times. The shortbread hardness was recorded as the force required to cut/break the biscuit using a texture analyzer (model TVT 6700, Perten Instruments Italia S.R.L., Rome, Italy) equipped with the P-BP70A probe. The peak force (N) at the breaking point represented the hardness of the shortbread. The shortbreads were rearranged and restacked to obtain the average measurements. Ten measurements per replication were taken and average values are reported.

#### 2.2.5. Spread Factor and Weight Loss Determination

The width (W) and thickness (T) of ten shortbreads were measured using a caliper after baking, and the spread factor was calculated by dividing these two values, according to the AACC, method 10–50.05 [17]. The weight of one hundred shortbreads before and after cooking was determined to calculate the weight loss by the difference [12]. 

#### 2.2.6. Sensory Analysis

In total, two sensory trials were carried out, discrimination and quantitative descriptive analysis. For each test, panelists were presented with two shortbreads in polyethylene bags, a glass of water to cleanse the palate, and the scorecard.

##### Sensory Discrimination and Preference Test

Thirty untrained panelists, aged from 20 to 50, of whom 14 were female and 16 were male, performed the triangle-discriminating test. The panelists were recruited by voluntary participation in the sensory analysis and asked to sign a consent form. Two tests were carried out, one comparing Q60S with Q100S and the other comparing Q60S with Q40S. The Evaluation Sheet is available in the Appendix A.

##### Quantitative Descriptive Analysis

The sensory evaluation was carried out over 2 days in 2 sessions in which the CS was compared with Q60S. Each panelist performed one replication of each treatment. Sixteen trained panelists, aged from 20 to 50, consisting of eight males and eight females, evaluated the shortbreads using the quantitative descriptive analysis technique. The panelists were recruited from the University personnel, asked to sign a consent form, and trained in 10 sessions to identify and determine descriptors relating to smell, taste, and texture. The descriptors and their corresponding definitions were available to the panelists during all sessions. Each panelist performed one replication of each treatment. A 10-point intensity scale was used wherein appearance was scored for surface appearance (uniformity), color, and global appearance; odor was scored for sweetness, vanilla, toasted, buttery, and overall odor; the taste was scored for bitter, buttery, vanilla, floury, sweetness, saltiness, eggy, and overall taste; consistency was scored for crispness, hardness, dryness, and pastiness. The Evaluation Sheet is available in the Appendix A.

#### 2.2.7. Statistical Analysis

A SigmaPlot software (15th version, SPSS Inc., Chicago, IL, USA) version 12.5 was used for the statistical analysis. A one-way analysis of variance (ANOVA), with Tukey’s post hoc test (*p* ≤ 0.05), was used to individuate significant variations between the different samples.

## 3. Results and Discussion

### 3.1. Chemical Quality Parameters of Flour

The quality parameters of WF, QF, and RF are compared in Table 2. WF was used for the formulation of traditional shortbreads, used as a control sample, and QF and RF were used, at different percentages, for the production of GF experimental shortbreads. The comparison underlines the different composition and quality parameters of the flours, as reported in the literature [18,19]. The QF had significantly (*p* < 0.05) higher ash, proteins, fat, and TDF levels than WF and RF. In particular, if compared with WF, QF had about 5, 1.5, 7, and 4 times higher content of ash, proteins, fat, and TDF, respectively. The content of carbohydrates, because of the higher amount of other parameters, in QF was significantly (*p* < 0.05) lower than in WF. If compared with RF, QF had about 7, 1.7, 17.5, and 2 times higher content of ash, proteins, fat, and TDF, respectively. WF and RF had comparable compositions, with similar carbohydrates, slightly higher proteins, higher fat, and a slightly higher ash content in WF; RF, on the other hand, showed a higher TDF content. The higher fat and sugar content of QF was not expected as a problem in the formulation of shortbreads and in their final composition, because in the recipe, the fat fraction was represented mainly by butter and the sugar fraction by sucrose.

Concerning antioxidant compounds and antioxidant capacity, QF had a level of total polyphenols twice higher compared with RF and 10 times higher compared with WF; the antioxidant activity, measured by the mean of the three different chosen assays, was the highest in QF, intermediate values were measured for RF, and the lowest values were verified for WF. The preliminary results on different flours underlined and confirmed the interesting composition and nutritional properties of QF, which is rich in valuable compounds such as proteins (providing energy, essential for many biological activities, good health maintenance, and the growth of the body), TDF, and polyphenols (with beneficial effects on human health) [14,20]. Moreover, QF showed the highest antioxidant activity for all three different assays.

Flour is the most important raw material for bakery products. Despite the poor technological properties of their proteins, GF flours have a main advantage in their nutritional properties and the well-balanced composition of the AA profile. AAs are key compounds, indispensable for many biochemical reactions. Free AAs, in particular, are the building blocks of protein molecules, and in food, these compounds contribute to flavor and taste [20]. They take part in the Maillard reaction (non-enzymatic browning), reacting with reducing sugars under high-temperature conditions, developing flavor and taste active compounds [10,12]. However, the Maillard reaction also gives rise to some negative molecules such as acrylamide; asparagine mainly but also other free AAs produce acrylamide by reacting with glucose and fructose at high temperatures [20]. Table 3 shows the AA composition of WF, QF, and RF. QF showed a significantly (*p* < 0.05) higher content of free AAs, which was 8 and 13 times higher than that of WF and RF, respectively. Moreover, QF had a significantly (*p* < 0.05) higher content of all essential AAs if compared with RF and WF.

Finally, for those concerned about the impact on the flavor and taste, QF had the highest number of bitterness AAs, namely, arginine, histidine, tyrosine, leucine, valine, methionine, isoleucine, phenylalanine, and lysine, accounting for 46% of the total, and the lowest number of sweetness AAs, namely, serine, alanine, glycine, and threonine, accounting for 11%. Both WF and QF had neutral AAs as the most representative fraction, while the least represented was the bitterness fraction in RF and the sweetness fraction in WF. Finally, if compared with WF, QF had a 1.7 times lower amount of asparagine, which is the main amino acid responsible for acrylamide production [20]. 

Regarding the fatty acid (FA) profile of different flours (Table 4), QF had a higher content of unsaturated FAs (88.8%), characterized by high amounts of linoleic, oleic, γ-linolenic, and α-linolenic acids, the last two in higher concentration than in WF and RF. This enhanced the nutritional quality of QF, because unsaturated FAs are beneficial for human health, as they can help to regulate blood cholesterol levels, play an anti-inflammatory role, and have a positive effect on the cardiocirculatory system [21].

From the obtained results, from a nutritional point of view, QF was found to be an excellent flour for the production of GF shortbreads due to its high TDF and protein content, interesting polyphenol content, and antioxidant activity. Moreover, QF had a significantly (*p* < 0.05) high content of free AAs, with an excellent presence of essential ones, and its fatty matrix was characterized by the presence of unsaturated fatty acids. It is also necessary to consider the presence of bitterness AAs and the possibility of related bitterness flavor problems in the finished shortbreads. Nevertheless, the experimental GF shortbreads formulated with the highest QF content could theoretically present the best nutritional quality. However, considering the poor textural properties of GF flours, particular attention was paid to the evaluation of the textural characteristics of the GF shortbreads, formulated according to the recipes reported in Table 1, that were analyzed in terms of hardness. In addition, the shortbreads with QF had a significantly (*p* < 0.05) high amount of AAs related to bitterness and this can cause flavor problems in the panelists’ compliance with the finished shortbreads, as well as greater hardness as highlighted by the textural analysis.

### 3.2. Control and Experimental Shortbread Textural Analysis

The results of the hardness determination on control and experimental shortbreads are reported in Figure 2 and are the mean of ten replications. The shortbreads most similar to the conventional ones (made up of 100% WF) were those formulated with 60% and 100% QF. Q40S showed an intermediate value of hardness. The shortbreads with the same hardness as the CS were selected and subjected to a discrimination test to verify the real differences between the shortbreads, and identify the one the panelists preferred. 

### 3.3. Sensory Discrimination and Preference between Q40S, Q60S, and Q100S

In the first trial, Q60S and Q100S were compared, to verify the real differences and identify the panelists’ preferred one between the two samples evaluated as the ones with the same hardness as the CS. The difference was underlined by 100% of panelists, and 90% of them expressed their preference for Q60S. Moreover, considering the potentially negative impact of the presence of QF on the flavor, it was checked whether Q40S (with an intermediate value of hardness) could be preferred over Q60S. The results showed that the difference was underlined by 70% of panelists, and 71% of them preferred Q60S. Q60S was selected as the experimental GF shortbread with the best textural and sensorial characteristics among the different formulations evaluated. These experimental shortbreads were evaluated for physicochemical parameters and sensorial attributes and were compared with the CS.

### 3.4. Quality Parameters of Selected Shortbreads

Table 5 shows the composition of the selected formulation of experimental GF shortbreads and CS. The values of moisture and aw, which influence the shelf life of shortbreads and their textural properties, were similar for the two samples [12]. The higher value of moisture found for Q60S could be explained by considering the significantly (*p* < 0.05) higher TDF content in quinoa shortbreads in comparison with the CS; in fact, the retention capacity of TDF is well-known [22]. The values of aw for both samples were below the threshold which assures microbiological stability and preservability [12]. Because of the different formulations and different quality parameters of the flours, GF shortbreads made up of QF and RF showed higher nutritional qualities than the CS, particularly about five times higher content of TDF and polyphenols, and significantly (*p* < 0.05) higher antioxidant activities, for all three different assays. According to Regulation (EC) No. 1924/2006, a food can be labeled as a “source of fiber” if it contains at least 3 g of fiber per 100 g of food [23]. Q60S fully accomplished the required level. Moreover, the GF experimental shortbreads had a fiber content very close to the claim “high in fiber”, which needs 6 g per 100 g. No significant differences were observed in the fat content of different shortbreads. The higher fat content of QF (7.0% vs. 1.01% of WF) did not influence the fat content of shortbreads, because of the presence of butter in both recipes. Moreover, a significantly (*p* < 0.05) lower content of carbohydrates was observed in Q60S. The lower carbohydrate content of Q60S produces an energy value that is definitively lower than the CS. Despite quinoa being a deeply investigated raw material for food production, it is difficult to find research studies about GF shortbreads to draw a comparison with our experimental product. In a recent paper concerning the use of 100% raw white quinoa flour for the formulation of shortbreads, the carbohydrate content was comparable (68.60%), while the protein and ash levels were higher (9.47% and 1.89%, respectively), probably because of the presence of rice in our product; the fat content was similar (14.63%), even if the used fat raw material was sunflower oil instead of butter. Finally, the energy value was also the same [24].

The ratio between the width and the thickness of the shortbreads allows us to calculate the spread factor, so higher values of this parameter indicate the lower thickness of the shortbreads with the same length and, therefore, worse leavening. Q60S had a spread factor of 62.50, significantly (*p* < 0.05) higher than CS (48.89), making the leavening of Q60S evidently worse, because of the absence of gluten. Furthermore, weight loss is due to the lowering of water content, which happens during the cooking phase due to evaporation. The analyzed samples showed similar values of weight loss.

Table 6 shows the AA composition of CS and Q60S; we can see how the GF shortbreads had a significantly (*p* < 0.05) higher amount of total free AAs (three times higher) and significantly (*p* < 0.05) higher amount of essential ones (two times higher) if compared with CS. Moreover, the fraction of bitterness AAs was significantly (*p* < 0.05) higher in Q60S and was the highest among the different flavor active fractions, together with the neutral one, while the sweetness fraction appeared to be the least abundant. This peculiar composition could probably influence the flavor of the quinoa GF shortbreads for the perception of bitterness typical of QF. In CS, the neutral fraction was the one most represented. 

Table 7 shows the FA composition of the two different shortbread samples. Q60S showed a higher percentage of unsaturated FAs, and in particular, polyunsaturated ones, compared with CS. Despite the presence of butter, quinoa seems to influence the FA composition of GF shortbreads. 

GF bakery products are usually nutritionally poor and are characterized by high amounts of fat and carbohydrates (due to the starchy ingredients), a low content of protein, and a high-energy value, with a poor intake of TDF [25]. In-depth clinical studies have highlighted the unbalanced intake of protein, carbohydrates, and fat along with the inadequate intake of the limiting nutrients TDF in celiacs as compared to people with a normal diet. In recent years, several studies have developed to ensure the nutritional quality, besides the technological quality, of GF bakery products [25,26]. The experimental GF shortbreads formulated in this research study are nutritionally valuable products, with an interesting intake of TDF, an interesting amount of essential AAs, and a balanced FA composition, despite the presence of butter in the recipe. 

### 3.5. Sensory Analysis of Selected Shortbreads

The obtained textural and analytical results underline the overall acceptable texture, comparable to that of CS, and the nutritional value of the GF experimental shortbread (Q60S). The use of QF as the main ingredient enables the improvement of the nutritional quality of GF shortbreads. The results of the sensory analysis (shown in Figure 3) allow for verifying if the consumers could appreciate the experimental shortbreads. Considering the overall components of visual and taste points of view, Q60S obtained slightly lower scores compared to CS. In detail, as far as the appearance is concerned, as regards surface uniformity and color, no significant differences (*p* < 0.05) were underlined, even if the global appearance scores of CS were significantly (*p* < 0.05) higher than those of Q60S, probably thanks to gluten. The overall odor obtained a similar score in both shortbread samples, even if the fat and vanilla odor were found as significantly higher in CS (*p* < 0.05). The odor of vanilla can be due to the wheat flour, while the fat odor is probably masked by the quinoa in Q60S, considering that the butter amount was the same. Considering the taste, a significantly higher bitter taste was observed in the quinoa shortbreads, while vanilla, floury, and overall taste obtained scores significantly lower (*p* < 0.05) than those of CS. Probably, the higher amount of bitterness AAs and phenolic compounds in QF caused the increased perception of bitter taste in the experimental GF shortbreads, together with the saponins that remained after the specific flour treatments [27,28]. Also, in this case, the bitter taste probably masked the fat, while the higher scores of vanilla and flour taste in CS were due to wheat; this led to a higher score for overall taste in CS. Interestingly, concerning consistency (one of the main goals to reach in the formulation of GF shortbreads), Q60S and CS have comparable scores in all the considered descriptors, namely, crispness, hardness, and pastiness, except for the statistically higher dryness (*p* < 0.05) in CS, as confirmed by the lower moisture value. The value of sensory hardness confirms the textural hardness value measured during the analysis of texture. 

## 4. Conclusions

In conclusion, this study clearly shows that the use of QF (at the amount of 60%) together with RF, in the formulation of GF shortbreads can be a valid tool for the production of nutritionally improved GF shortbreads. The use of QF and RF enables the enhancement of the nutritional value of shortbreads, with a significant decrement in carbohydrate content and, as a direct consequence, in energy value, and a significant increase in TDF content. 

QF in the recipe allows an increment in TDF of about five times and allows the experimental GF shortbreads to be labeled as a “source of fiber”, which can increase fecal mass and contribute to gut health. Moreover, the presence of QF in the formulation increases by about five times the polyphenol content, which contributes to an increase in the antioxidant capacity of shortbreads, with potential anticarcinogenic effects. Additionally, the experimental GF shortbreads were also rich in AAs, including essential ones, which are crucial for many biological activities, good health maintenance, and the growth of the body. 

Finally, the experimental shortbreads could meet consumer acceptance thanks to their high healthier composition and overall good sensory features and could attract consumers who want to purchase products with allergy/intolerance or social/ethical claims in order to make healthier and more sustainable food choices [29,30]. Furthermore, in future studies, the sensory characteristics could be further improved through the use, in recipes, of adjuvants such as hydrocolloids, broadly used in the food industry as functional ingredients because of their capacity to affect the textural and sensory properties of a foodstuff. Then, market acceptance can be checked using the consumer test. Moreover, the product can be improved, from a nutritional point of view, by replacing the butter with a fat raw material characterized by a higher percentage of unsaturated fatty acids. 

## Figures and Tables

**Figure 1 foods-13-00377-f001:**
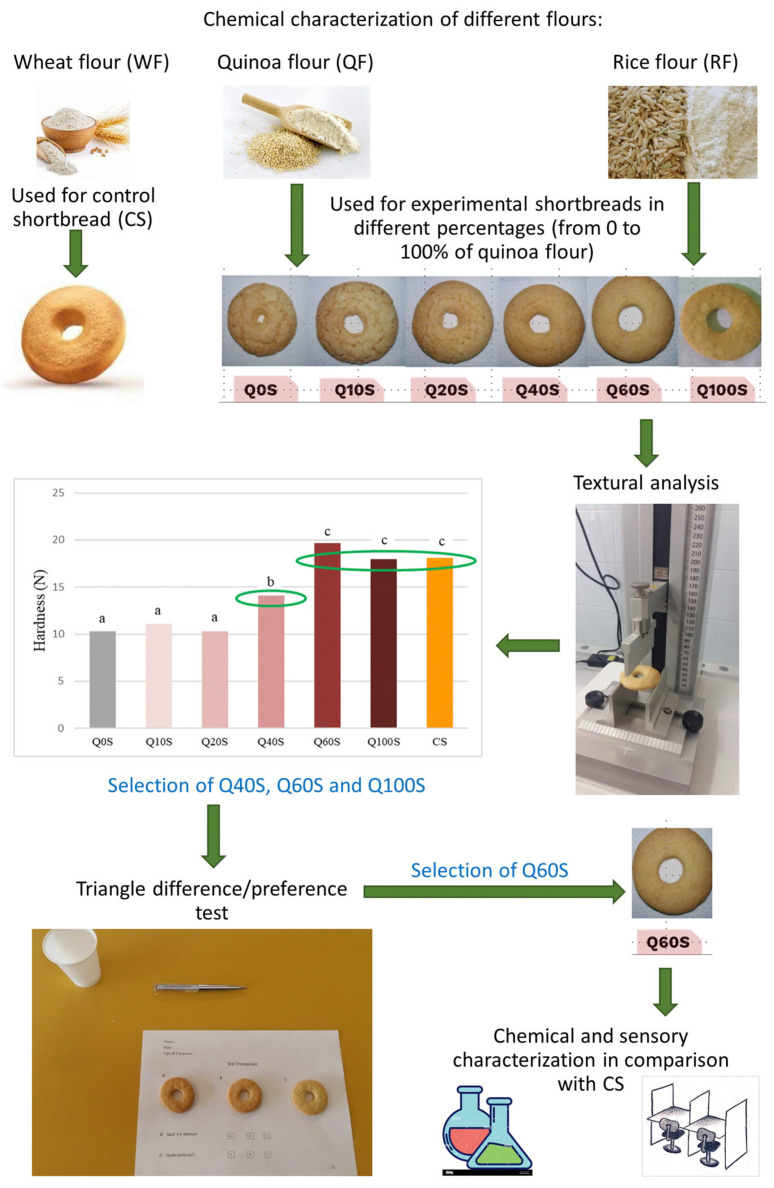
Experimental design of research study. Values expressed by histogram bars with different letters are significantly different (*p* ≤ 0.05). The green circle indicates the samples selected for the triangle-discriminating test.

**Figure 2 foods-13-00377-f002:**
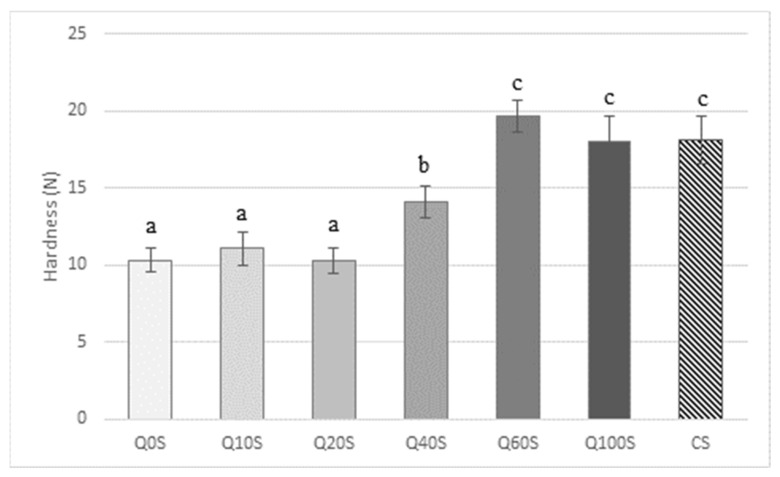
Textural properties of experimental and traditional shortbreads. *n* = 10; CS = control wheat shortbread; QnS: *n* = 0, 10, 20, 40, 60, 100% of quinoa flour; N = Newton. Bars with different letters are significantly different (*p* ≤ 0.05).

**Figure 3 foods-13-00377-f003:**
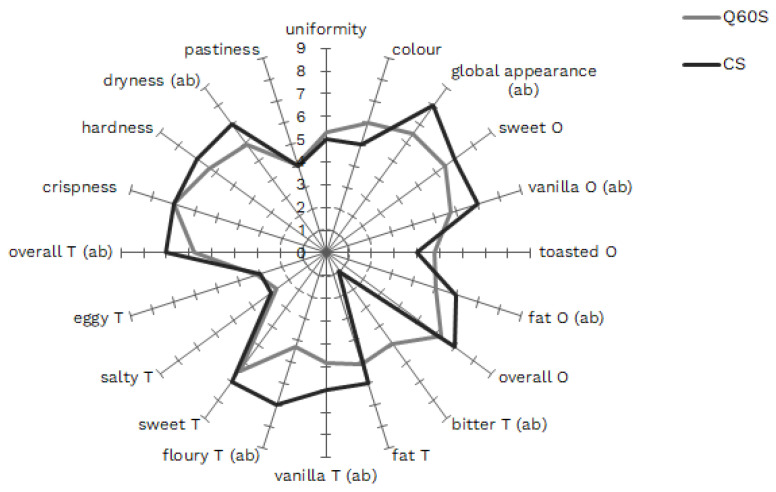
Sensory analysis of shortbreads. CS: traditional shortbread; Q60S: 60% quinoa flour shortbreads. O: odor; T: taste. Descriptors for which score values are significantly different (*p* ≤ 0.05) are labeled with different letters between brackets referring to samples CS and Q60S, respectively.

**Table 1 foods-13-00377-t001:** Shortbread recipes.

Ingredients (g)	CS	Q0S	Q10S	Q20S	Q40S	Q60S	Q100S
Wheat flour	325	-	-	-	-	-	-
Quinoa flour	-	-	34.5	69	138	207	345
Rice flour	-	345	310.5	276	207	138	-
Sucrose	180	180	180	180	180	180	180
NaCl	1.25	1.25	1.25	1.25	1.25	1.25	1.25
(NH_4_)HCO_3_	1	1	1	1	1	1	1
NaHCO_3_	1.50	1.50	1.50	1.50	1.50	1.50	1.50
Butter	100	100	100	100	100	100	100
Yolk	24	24	24	24	24	24	24
Egg white	37	37	37	37	37	37	37
Water	30	30	30	30	30	30	30

CS: control shortbread; Q0S, Q10S, Q20S, Q40S, Q60S, Q100S: shortbreads with 0, 10, 20, 40, 60, and 100% of quinoa flour, respectively.

**Table 2 foods-13-00377-t002:** Chemical quality parameters of flour samples.

Quality Parameter	WF	QF	RF
Moisture %	10.60 ± 0.25 a	9.22 ± 0.11 b	2.59 ± 0.02 c
Carbohydrates (% dm)	87.26 ± 0.38 a	68.9 ± 1.3 b	87.5 ± 0.5 a
Sugars (% dm)	1.00 ± 0.01 b	3.5 ± 0.3 a	nd
Ash (% dm)	0.57 ± 0.01 b	2.69 ± 0.01 a	0.40 ± 0.01 c
Proteins (% dm)	9.41 ± 0.08 b	13.8 ± 0.1 a	8.1 ± 0.1 c
Fat (% dm)	1.01 ± 0.01 b	7.0 ± 0.1 a	0.4 ± 0.1 c
TDF (% dm)	1.76 ± 0.33 c	7.7 ± 1.4 a	3.6 ± 0.5 b
aw	0.57 ± 0.01 a	0.44 ± 0.01 a	0.50 ± 0.01 a
Total Polyphenols (GA mg/g)	0.16 ± 0.01 c	1.64 ± 0.08 a	0.76 ± 0.04 b
ABTS (TE/g)	0.15 ± 0.03 c	0.69 ± 0.08 a	0.29 ± 0.01 b
DPPH (TE/g)	10.68 ± 1.00 c	75.77 ± 0.80 a	70.10 ± 0.91 b
FRAP (TE/g)	0.98 ± 0.03 c	3.58 ± 0.05 a	1.12 ± 0.04 b

*n* = 2 analytical replicates; data are shown as mean ± standard deviation; WF = wheat flour; QF = quinoa flour; RF = rice flour; dm = dry matter; TDF = total dietary fiber; values on the same row with different letters are significantly different (*p* ≤ 0.05).

**Table 3 foods-13-00377-t003:** Amino acid composition of flours.

AAs (mg/Kg dm)	WF	QF	RF
Aspartic Acid	105.8 ± 7.9 b^I^	228.7 ± 15.9 a^M^	17.6 ± 0.2 c^H^
Glutamic Acid	60.9 ± 0.9 b^G^	692.7 ± 0.7 a^O^	43.6 ± 1.2 c^L^
Asparagine	87.2 ± 0.3 a^H^	50.0 ± 3.2 b^DE^	52.8 ± 1.1 b^M^
Serine	11.8 ± 3.0 b^CD^	64.9 ± 11.4 a^FG^	8.7 ± 0.2 b^F^
Glutamine	44.5 ± 0.1 b^F^	163.1 ± 7.1 a^L^	6.9 ± 0.4 c^E^
Histidine	8.3 ± 0.6 b^ABC^	254.1 ± 0.8 a^N^	4.5 ± 0.1 c^D^
Arginine	25.4 ± 0.2 b^E^	736.1 ± 0.5 a^P^	6.3 ± 0.5 c^E^
Glycine	13.1 ± 0.9 b^D^	69.4 ± 8.2 a^G^	13.9 ± 0.2 b^G^
Alanine	29.0 ± 1.1 b^E^	146.4 ± 5.9 a^I^	26.2 ± 1.5 b^I^
Tyrosine	12.3 ± 0.5 b^CD^	56.1 ± 3.7 a^EF^	4.1 ± 0.4 b^CD^
*Threonine*	8.4 ± 0.7 b^ABC^	37.6 ± 4.6 a^ABC^	4.8 ± 0.1 c^D^
*Methionine*	4.8 ± 0.5 b^A^	27.6 ± 0.1 a^A^	3.3 ± 0.2 c^BC^
*Tryptophan*	63.8 ± 0.7 b^G^	74.4 ± 0.3 a^GH^	2.8 ± 0.1 c^B^
*Valine*	13.8 ± 1.2 b^D^	85.5 ± 0.3 a^H^	6.7 ± 0.1 c^E^
*Phenylalanine*	11.3 ± 0.5 b^CD^	45.2 ± 4.0 a^CDE^	2.9 ± 0.2 c^B^
*Isoleucine*	6.3 ± 0.3 b^AB^	28.8 ± 2.0 a^A^	1.4 ± 0.1 c^A^
*Leucine*	10.3 ± 0.3 b^BCD^	32.6 ± 2.4 a^AB^	2.6 ± 0.2 c^B^
*Lysine*	10.6 ± 1.0 b^BCD^	42.9 ± 4.4 a^BCD^	4.1 ± 0.3 c^CD^
Ʃ AAs	527.6 ± 16.3 b	2836.1 ± 40.7 a	213.1 ± 2.9 c
Ʃ sweetness AAs	62.2 ± 5.7 b^A^	318.3 ± 30.1 a^A^	53.6 ± 2.0 c^B^
Ʃ bitterness AAs	103.2 ± 5.1 b^B^	1308.9 ± 18.2 a^C^	35.8 ± 2.1 c^A^
Ʃ neutral AAs	365.2 ± 9.9 b^C^	1208.9 ± 27.2 a^B^	123.8 ± 3.9 c^C^

*n* = 2 analytical replicates; data are shown as mean ± standard deviation; AAs = amino acids; WF = wheat flour; QF = quinoa flour; RF = rice flour; dm = dry matter; the names of essential amino acids are written in italics. Values in the same row with different letters and values in the same column with different superscript letters are significantly different (*p* ≤ 0.05).

**Table 4 foods-13-00377-t004:** Fatty acid composition of gluten-free flours.

Fatty Acids (% dm)	WF	QF	RF
Butyric C4:0	0.21 ± 0.02 a^C^	nd	nd
Myristic C14:0	0.13 ± 0.01 b^A^	0.12 ± 0.01 b^B^	0.58 ± 0.01 a^C^
Palmitic C16:0	17.14 ± 0.01 a^H^	8.72 ± 0.02 b^I^	16.72 ± 0.01 a^H^
Palmitoleic C16:1 ω7	0.14 ± 0.01 c^AB^	0.05 ± 0.01 a^A^	0.22± 0.01 b^A^
Stearic C18:0	1.17 ± 0.01 b^E^	0.79 ± 0.01 ^E^	2.25± 0.01 b^G^
Oleic C18:1 ω9	15.83 ± 0.01 c^G^	29.75 ± 0.02 b^L^	41.16 ± 0.01 a^L^
Linoleic C18:2 ω6	60.72 ± 0.08 ^I^	47.65 ± 0.03 b^M^	34.88 ± 0.01 c^I^
γ-linolenic C18:3 ω6	3.30 ± 0.02 b^F^	7.57 ± 0.01 b^H^	1.29 ± 0.01 c^F^
Eicosenoic C20:1 ω9	0.19 ± 0.01 c^BC^	0.59 ± 0.01 b^D^	0.89 ± 0.02 a^E^
α-linolenic C18:3 ω3	0.74 ± 0.01 b^D^	1.80 ± 0.01 ^G^	0.70 ± 0.01 b^D^
Arachidonic C20:4 ω6	0.21 ± 0.01 c^C^	0.78 ± 0.01 a^E^	0.42 ± 0.01 b^B^
Trycosilic C23:0	nd	1.58 ± 0.05 ^F^	nd
Eicosapentaenoic C20:5 ω3	0.22 ± 0.01 b^C^	0.30 ± 0.01 b^C^	0.89 ± 0.01 a^E^
Nervonic C24:1 ω9	nd	0.30 ± 0.01 a^C^	nd
Ʃ Saturated	18.65 ± 0.05 ^B^	11.21 ± 0.09 b^A^	19.55 ± 0.08 a^A^
Ʃ Unsaturated	81.35 ± 0.16 b^D^	88.79± 0.12 a^D^	80.45 ± 0.01 b^D^
Ʃ Monounsaturated	16.16 ± 0.03 c^A^	30.69 ± 0.05 b^B^	42.27 ± 0.03 a^C^
Ʃ Polyunsaturated	65.19 ± 0.13 ^C^	58.10 ± 0.07 b^C^	38.18 ± 0.05 c^B^

*n* = 2 analytical replicates; data are shown as mean ± standard deviation; WF = wheat flour; QF = quinoa flour; RF = rice flour; dm = dry matter; nd = not detectable. Values in the same row with different letters and values in the same column with different superscript letters are significantly different (*p* ≤ 0.05).

**Table 5 foods-13-00377-t005:** Chemical quality parameters of the control and 60% quinoa flour shortbreads.

Quality Parameter		CS	Q60S
Energy value	(kcal/100 g)	472	446
	(kJ/100 g)	1975	1866
Moisture (% dm)		1.56 ± 0.02 b	2.51 ± 0.09 a
Carbohydrates (% dm)		73.28 ± 0.13 a	67.30 ± 1.30 b
Sugars (% dm)		24.77 ± 0.22 a	25.10 ± 0.10 a
Ash (% dm)		1.01 ± 0.05 a	1.09 ± 0.03 a
Proteins (% dm)		7.13 ± 0.19 a	7.60 ± 0.10 a
Fat (% dm)		15.99 ± 0.17 a	16.30 ± 0.60 a
TDF (% dm)		1.03 ± 0.02 b	5.20 ± 0.50 a
a_w_		0.28 ±0.01 a	0.36 ± 0.05 a
Spread factor		48.89 ±0.82 b	62.50 ± 1.08 a
Weight loss		1.67 ± 0.03 a	1.33 ± 0.03 b
Weight loss (%)		16.06 ± 0.23 a	13.40 ± 0.20 b
T-Polyphenols (GA mg/g)		1.25 ± 0.01 b	5.73 ± 0.01 a
ABTS (TE/g)		0.10 ± 0.01 b	0.37 ± 0.06 a
DPPH (TE/g)		25.07 ± 0.12 b	70.03 ± 0.24 a
FRAP (TE/g)		2.77 ± 0.03 b	8.37 ± 0.09 a

*n* = 2 analytical replicates; data shown as mean ± standard deviation; CS = control wheat shortbread; Q60S = 60% quinoa shortbreads; dm = dry matter; a_w_ = water activity; TDF = total dietary fiber; GA = gallic acid; TE = Trolox equivalents; values in the same row with different letters are significantly different (*p* ≤ 0.05).

**Table 6 foods-13-00377-t006:** Amino acid composition of control and 60% quinoa flour shortbreads.

AAs (mg/Kg dm)	CS	Q60S
Aspartic Acid	44.8 ± 0.1 b^O^	82.3 ± 6.0 a^L^
Glutamic Acid	32.5 ± 0.6 b^M^	239.0 ± 11.9 a^N^
Asparagine	40.0 ± 0.3 a^N^	31.7 ± 1.7 b^EFG^
Serine	9.6 ± 0.5 b^F^	30.0 ± 3.4 a^DEF^
Glutamine	22.1 ± 0.2 b^H^	40.2 ± 1.8 a^G^
Histidine	5.1 ± 0.3 b^C^	72.0 ± 3.4 a^I^
Arginine	26.9 ± 0.1 b^I^	217.8 ± 12.2 a^M^
Glycine	7.3 ± 0.1 b^D^	29.7 ± 0.9 a^DEF^
Alanine	28.0 ± 0.5 b^L^	59.6 ± 3.9 a^H^
Tyrosine	9.1 ± 0.5 b^F^	28.6 ± 1.8 a^CDEF^
*Threonine*	5.3 ± 0.2 b^C^	23.7 ± 2.6 a^BCDE^
*Methionine*	2.7 ± 0.1 b^A^	10.5 ± 0.1 a^A^
*Tryptophan*	21.6 ± 0.2 a^H^	21.0 ± 0.9 a^BCD^
*Valine*	13.7 ± 0.3 b^G^	35.4 ± 1.9 a^FG^
*Phenylalanine*	8.2 ± 0.3 b^E^	19.0 ± 0.6 a^ABC^
*Isoleucine*	7.1 ± 0.2 b^D^	14.5 ± 0.7 a^AB^
*Leucine*	9.7 ± 0.3 b^F^	21.9 ± 1.3 a^BCD^
*Lysine*	4.0 ± 0.2 b^B^	23.4 ± 0.7 a^BCDE^
Ʃ AAs	297.7 ± 5.0 b	1000.3 ± 55.8 a
Ʃ sweetness AAs	50.2 ± 1.3 b^A^	143.0 ± 10.8 a^A^
Ʃ bitterness AAs	86.5 ± 2.3 b^B^	443.1 ± 22.7 a^B^
Ʃ neutral AAs	161.0 ± 1.4 b^C^	414.2 ± 22.3 a^B^

*n* = 2 analytical replicates; data are shown as mean ± standard deviation; AAs = amino acids; CS = control wheat shortbread; Q60S = 60% quinoa shortbreads; dm = dry matter; the names of essential amino acids are written in italics. Values in the same row with different letters and values in the same column with different superscript letters are significantly different (*p* ≤ 0.05).

**Table 7 foods-13-00377-t007:** Fatty acid composition of control and 60% quinoa flour shortbreads.

Fatty Acids (% dm)	CS	Q60S
Butyric C4:0	2.13 ± 0.03 a^I^	2.01 ± 0.05 a^H^
Caproic C6:0	1.66 ± 0.01 a^G^	1.64 ± 0.01 a^G^
Caprylic C8:0	1.13 ± 0.01 a^F^	1.10 ± 0.01 a^F^
Capric C10:0	2.69 ± 0.01 a^L^	2.62 ± 0.01 a^I^
Lauric C12:0	3.27 ± 0.01 a^M^	3.17 ± 0.02 b^L^
Myristic C14:0	12.01 ± 0.04 a^P^	10.12 ± 0.08 b^O^
Myristoleic C14:1 ω9	0.85 ± 0.01 b^E^	0.94 ± 0.01 a^E^
Pentadecanoic C15:0	0.08 ± 0.01 b^A^	1.15 ± 0.01 a^F^
Palmitic C16:0	31.08 ± 0.02 a^R^	29.51 ± 0.01 b^Q^
Palmitoleic C16:1 ω7	1.79 ± 0.01 a^H^	1.57 ± 0.02 b^G^
Heptadecanoic C17:0	0.1 ± 0.01 b^A^	0.58 ± 0.01 a^C^
Heptadecenoic C17:1	0.1 ± 0.01 b^A^	0.24 ± 0.01 a^B^
Stearic C18:0	10.37 ± 0.03 a^O^	9.16 ± 0.04 b^N^
Oleic C18:1 ω9	25.55 ± 0.04 a^Q^	25.15 ± 0.13 a^P^
Linoleic C18:2 ω6	5.14 ± 0.03 b^N^	8.36 ± 0.02 a^M^
γ-linolenic C18:3 ω6	0.1 ± 0.01 b^A^	1.17 ± 0.01 a^F^
Eicosenoic C20:1 ω9	0.65 ± 0.01 a^C^	0.12 ± 0.01 b^A^
α-linolenic C18:3 ω3	0.77 ± 0.01 b^D^	0.85 ± 0.01 a^D^
Behenic C22:0	0.10 ± 0.01 a^A^	0.11 ± 0.01 a^A^
Eicosatrienoic C20:3 ω6	nd	0.11 ± 0.01 ^A^
Arachidonic C20:4 ω6	0.2 ± 0.01 a^B^	0.13 ± 0.01 b^A^
Trycosilic C23:0	0.23 ± 0.01 a^B^	0.22 ± 0.04 a^B^
Ʃ Saturated	64.95 ± 0,20 a^A^	61.36 ± 0,30 b^D^
Ʃ Unsaturated	35.15± 0,14 b^B^	38.64 ± 0,23 a^C^
Ʃ Monounsaturated	28.94 ± 0,08 a^C^	28.02 ± 0,17 b^B^
Ʃ Polyunsaturated	6.21 ± 0.06 b^D^	10.62 ± 0.06 a^A^

*n* = 2 analytical replicates; data are shown as mean ± standard deviation; CS = control wheat shortbread; Q60S = 60% quinoa shortbreads; dm = dry matter; nd = not detectable. Values in the same row with different letters and values in the same column with different superscript letters are significantly different (*p* ≤ 0.05).

## Data Availability

The data presented in this study are available on request from the corresponding author. The data are not publicly available because are designed to be used in other ongoing research and should be protected before official publication.

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
