# Peer review of "Quinoa (Chenopodium Quinoa Willd.) as Functional Ingredient for the Formulation of Gluten-Free Shortbreads"

_foods, 2024, doi:10.3390/foods13030377_

Round 1

Reviewer 1 Report

Comments and Suggestions for Authors

Quinoa (Chenopodium quinoa Willd.) as functional ingredient for the formulation of gluten-free shortbreads

The study provides an extensive comparison of quinoa flour (QF) against wheat flour (WF) and rice flour (RF) in terms of nutritional composition and its application in shortbread formulations. The comprehensive evaluation of chemical parameters, amino acids, fatty acids, and sensory characteristics offers valuable insights into the potential benefits of incorporating QF into gluten-free formulations. This research suggests that incorporating QF into gluten-free baked goods could offer consumers healthier options without compromising sensory appeal. The study opens opportunities for producing nutritionally enriched gluten-free products that meet consumer demands for healthier and more sustainable food choices.

General comment:

However, there are a few areas that require clarification. The methodology section lacks detailed information on the sourcing and preparation of the flours used in the study, which could potentially impact the observed differences in composition. Additionally, there are minor inconsistencies in the presentation of data across tables that need addressing, particularly in the amino acids and fatty acids composition sections. Furthermore, while the study admirably highlights the nutritional advantages of QF, there is a need for a more robust sensory analysis, including a larger sample size and diverse consumer panels, to strengthen the conclusions regarding the preference and acceptability of the GF shortbreads made with QF.

In the manuscript, there's a reference to 'rheological properties,' but it's unclear which specific rheological properties are being alluded to. Therefore, it's recommended to either incorporate rheological analyses of the flour and dough or use an appropriate term. If 'hardness' is the sole property considered among rheological properties, then 'textural property' would be a more appropriate term.

Specific comments:

The manuscript's structure needs reorganization to ensure a coherent flow from materials and methods to conclusions. Aligning the sequence of methods, results, and discussion will enhance the manuscript's cohesion and reader comprehension.

Consistency in terminology is advised; use either 'shortbread' or 'biscuits,' avoiding the use of both terms interchangeably.

In the abstract, there is 'TDF' abbreviation without explanation. It's essential to check the manuscript for similar cases and ensure clarity throughout the document.

Materials and Methods: The statement 'All samples were analyzed in duplicate' lacks clarification regarding whether the samples were prepared in one or multiple batches. Further explanation is needed for clarity.

·         Detailed explanations of the analytical techniques employed (e.g., chemical composition, sensory evaluation) would enhance reproducibility. Rather than stating, for instance, 'An HPLC-ELSD system was used for the determination of free sugar content as reported by Sileoni et al. [11],' provide specific methods and equipment details.

·         The description of the sensory analysis (section '2.2.4. Sensory analysis') requires additional elaboration, including details about Ethics Committee approval, the selection process for sensory experts, their consent, and other essential procedural information. The addition of an Evaluation Sheet in the Supplement containing questions for sensory experts would be beneficial.

Results and Discussion: The paper demonstrates a meticulous analysis of the chemical composition of different flours and their resultant shortbreads, presenting comprehensive data and insightful discussions. With minor refinements and additional contextualization, the manuscript can further elevate its contribution to the field of nutritional science and gluten-free baking. Suggestions for Improvement:

·         Further Comparative Analysis: While the paper excellently compares the parameters among the flours and shortbreads, a deeper comparative analysis across multiple studies or against established nutritional standards could fortify the claims made regarding the nutritional quality of the experimental shortbreads.

·         Clarification on Sensory Attributes: The discussion touches upon the potential impact of amino acid fractions on taste perception, but further elaboration on sensory attributes based on the findings would enhance the understanding of the potential taste and flavor differences between the control and experimental shortbreads.

·         Inclusion of Statistical Significance: Emphasizing the statistical significance of differences observed in the parameters between the control and experimental samples could reinforce the credibility of the conclusions drawn.

·         Figure 1 should include standard deviation values, and the y-axis should indicate the specific parameter. Additionally, 'hardness' is more accurately termed a textural property, necessitating a correction in the figure's title.

Conclusion: The conclusion is well-structured, encompassing the key findings and suggesting future directions for research to enhance the nutritional and sensory aspects of GF shortbreads formulated with QF and RF. However, some aspects could benefit from further elaboration. For instance, while it highlights the nutritional enhancements, it could provide more specific details or data supporting the claims of increased polyphenol content and the impact on essential amino acids (AAs). Additionally, a more detailed discussion on the methodology used to evaluate sensory characteristics and consumer compliance could strengthen the conclusion.

Reviewer 2 Report

Comments and Suggestions for Authors

The manuscript is addressing the escalating prevalence of celiac disease and gluten intolerance and it is focused on enhancing gluten-free shortbreads, the study employs quinoa flour in various formulations to strike a balance between improved nutritional composition and maintaining an acceptable sensory profile. The manuscript is well-prepared, but it contains a significant number of minor and major shortcomings.

In the abstract, it is necessary to improve the sentence in lines 18-20 for a more logical expression.

Overall Editing:

        Edit the entire manuscript for formatting, including justification, paragraph indentation, and proper use of sections. Clarify and explain abbreviations when first used and additioanlly when they appear in the abstract.

Lines 83-98:

Instead of describing  the methods and results, provide a more detailed description of the research objective and hypothesis.

Section 1.1 - Experimental Design:

Simplify the explanation of the experimental design, potentially using a graphical scheme for better clarity.

Include information about the manufacturers of devices used (e.g., mixer, spectrophotometer), specifying their parameters if available.

Expand on the methods section for a more comprehensive understanding.

Clarify abbreviations in the materials section.

Specify which statistical test was employed for testing homogeneous groups (e.g., Duncan, Tukey).

Expand a discussion section and provide notes for better understanding of Tables 3 and 4.

Figure 1: Change the caption from "rheological properties" to focus specifically on hardness.

Explain the reason for the different colors used in parts of the amino acid tables.

Round 2

Reviewer 1 Report

Comments and Suggestions for Authors

The authors have embraced the majority of suggestions provided by the reviewers and improved the manuscript; however, they overlooked making the following correction: Consistency in terminology is advised; use either 'shortbread' or 'biscuits,' avoiding the use of both terms interchangeably.

Reviewer 2 Report

Comments and Suggestions for Authors

The authors responded to all reviewer comments and the manuscript was significantly improved.
